# Anthropometric analysis of orbital and nasal parameters for sexual dimorphism: New anatomical evidences in the field of personal identification through a retrospective observational study

**Roberto Scendoni[1], Jeta Kelmendi[2], Isabella Lima Arrais Ribeiro[3], Mariano Cingolani[1], Francesco De Micco[4,5]\*, Roberto Cameriere[6,7]**

1 Department of Law, Institute of Legal Medicine (AgEstimation Project), University of Macerata, Macerata, Italy, 2 Faculty of Medicine, Department of Orthodontic, University of Prishtina, Alma Mater Europaea, Campus Rezonanca, Prishtina, Kosovo, 3 Postgraduate Program in Dentistry, Federal University of Paraíba, Campus I, João Pessoa, Brazil, 4 Bioethics and Humanities Research Unit, Campus Bio-Medico University of Rome, Rome, Italy, 5 Department of Clinical Affairs, Campus Bio-Medico University Hospital Foundation, Rome, Italy, 6 Department of Forensic Medicine (AgEstimation Project), University of Molise, Campobasso, Italy, 7 Department of Forensic Medicine, I.M. Sechenov First Moscow State Medical University, Moscow, Russia

\* f.demicco@policlinicocampus.it

## Abstract

Orbital and nasal parameters among modern humans show considerable variation, which affects facial shape, and these characteristics vary according to race, region, and period in evolution. The aim of the study was to ascertain whether there are sex differences in the orbital and/or nasal indexes and/or the single measurements used to calculate these in a Kosovar population. The following parameters were taken into consideration: orbital height (OH), orbital width (OW), nasal height (NH), and nasal width (NW). The ratios between orbital index/nasal index (RONI) were calculated. All measurements were obtained from a population sample comprising 408 individuals. The accuracy in sex prediction was 52.86% (CI95% = 45.05%–60.67%) for NW and 64.96% for NH (CI95% = 57.50%– 72.42%). The difference between male and female indexes was statistically significant (P < 0.05). The anthropometric study revealed that only NW and NH are configured as predictors of sexual dimorphism. It could be useful to increase the number of samples to test the discriminant function in other population groups.

## Introduction

Ethnicity assessment and identification of skeletal remains are important and difficult tasks for forensic scientists with significant ethical and legal implications [1]. Numerous methods using craniofacial measurements and indexes are available and useful for these purposes [2–4]. The manner of determining the parameters needed for the estimation of these directories depends

**Data Availability Statement:** All relevant data are within the paper and its Supporting information files.

**Funding:** The author(s) received no specific funding for this work.

**Competing interests:** The authors have declared that no competing interests exist.

on the sort of samples used [5]. Extensive research has been conducted worldwide on the dimensions and volumes of the orbits and nasal cavity. Calipers are still commonly used as linear measuring instruments to measure the dimensions of the orbital and nasal margins [6,7]. However, advanced radiological techniques such as computed tomography (CT) scanners have proven to be a better choice since they facilitate sample collection [6]. The orbital index (OI) and nasal index (NI) are calculated using measurements between several landmarks in the respective facial cavity [7]. Orbital and nasal cavity dimensions and volumes, in various populations, have been investigated by many authors [8–10], and different populations have been found to exhibit considerable variation in these values [11–14]. Indeed, population-based variations are predictable results of evolutionary processes; natural selection acts on inheritable mutations, which may relate to current environmental pressures, genetic implication, past and present hybridization between geographically distinct populations, and the present selective adaptation of human varieties to their environment [15].

Overall, these studies have indicated some level of ethnic and racial variation in the OI and NI of various population groups [16–19]. In modern human groups, the appearance of the orbital and nasal cavities varies considerably [9]. Craniofacial morphometric analysis is conducted using OI and NI parameters, based on the ratio of the orbital/nasal height to its width multiplied by 100. These ratios affect the shape of the face and vary with race, regions within the same race, and periods in evolution [20–23].

However, to date there has been no morphometric study on OI and NI and their relationship to biological sex prediction in the Kosovar population. Therefore, this study of the orbital and nasal morphometry in the skulls of a Kosovar population using a CT scanner represents an important contribution to the literature which will expand knowledge in various fields, such as forensic medicine, and help to explore trends in evolutionary and ethnic differences, among different races in particular, where forensic data is not available. Thus, the main objective of this research is to ascertain whether there are sex differences in the orbital and/or nasal indexes and/or the single measures used to calculate these. The authors assume that there may be differences between the two sexes at least in one or more parameters necessary to calculate either the nasal index or the orthotic index in the population under examination. If the hypothesis is confirmed, on the basis of the results obtained, the authors will provide a plausible explanation in this regard.

## Materials and methods

The study was approved by the Ethical Issues Committee of the Kosovo Chamber of Dentists (approval no. 27 of 18 May 2022). To facilitate the evaluation of strengths and limitations and the generalisability of the results, the observational study was developed according to the STrengthening the Reporting of Observational studies in Epidemiology (STROBE) [24]. A retrospective study design was used in this research involving people living in Kosovo. The CT scans were anonymously extracted and limited data were extracted that were strictly necessary and relevant for the conduct of the study. The extracted data are all available in the manuscript. CT examinations were analyzed from 1 February to 15 March 2022. The inclusion criteria were males and females of Kosovar ethnicity, over the age of 18, who underwent CT scan to investigate neurological symptoms. Exclusion criteria included growth diseases, endocrine disorders or osteodystrophy, previous fractures of the of the skull and the facial massif. Finally, all blurred CT images were disregarded. A sample of 408 individuals was randomly divided into two subsamples, one constituting approximately 2/3 (n = 251) of the initial sample, and the other containing approximately 1/3 of the initial sample (n = 157). This division was important for the steps of training (2/3 of sample) and testing (validation) (1/3 of sample) variables of

interest for the prediction of sex. Therefore, 251 individuals were evaluated, 110 females and 141 males, with a mean age of 37.41 (±20.56) and 33.87 (±19.58), respectively. The random splitting of the sub-samples could represent a potential research bias. The sample size of the present study was calculated based on the effects observed in the study by Kotian et al. [25]. From this study, the mean effect size of difference between groups of 0.20 (Cohen's d) was observed. Adopting a 5% type I error (95% study confidence level), a 20% type II error (80% study power), the effect size observed in the mentioned study and, considering the comparison tests of the groups of the bilateral type (two-sided), it was calculated that at least 394 sample units were necessary for the accomplishment of the study. As they are available, we conducted the study with 408 sample units. The calculations were performed using the G*Power statistical software.

The study was carried out on frontal views of head CT scans performed on patients who had attended radiology units (private and public) in various cities in Kosovo. The height and width of the orbital cavity and nasal aperture were measured on the CT scan of the frontal bone using a digital ruler (in mm). The following parameters were measured: orbital height (OH) and nasal height (NH)–the maximum distance between the upper and lower margins of the orbital cavity rim and nasal aperture rim (Fig 1); orbital width (OW) and nasal width (NW)–the distance from the midpoint of the medial margin of the orbital/nasal aperture to the midpoint of the lateral margin of the same orbit/nasal aperture. The mean orbital and nasal indexes were calculated by the formula: height/width x 100. Only normal CT scans were used (cases with bone pathology and fractures were excluded from the study). Finally, the ratios between orbital index/nasal index (RONI) were calculated.

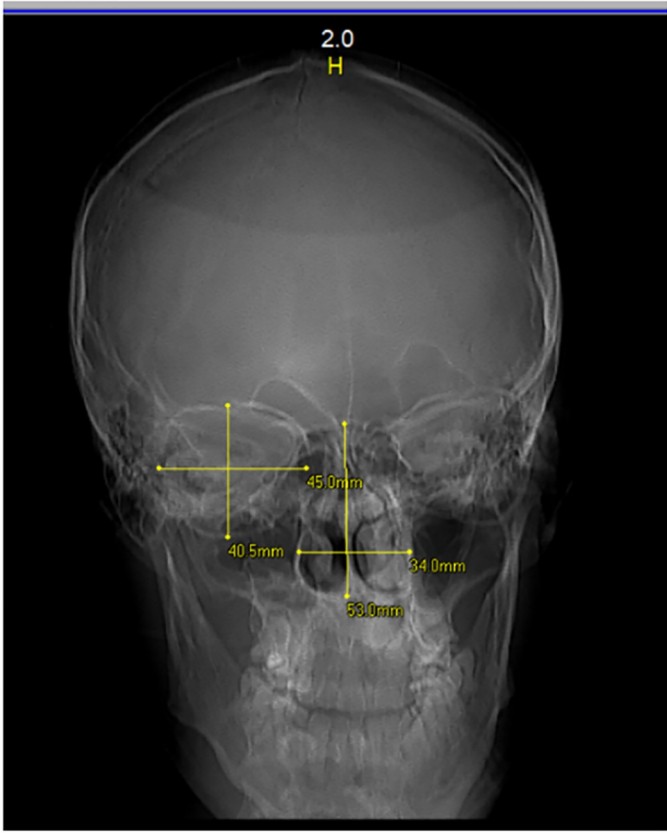

**Fig 1. Plain film showing the orbital and nasal margins and measurements of orbital/nasal width and height.**

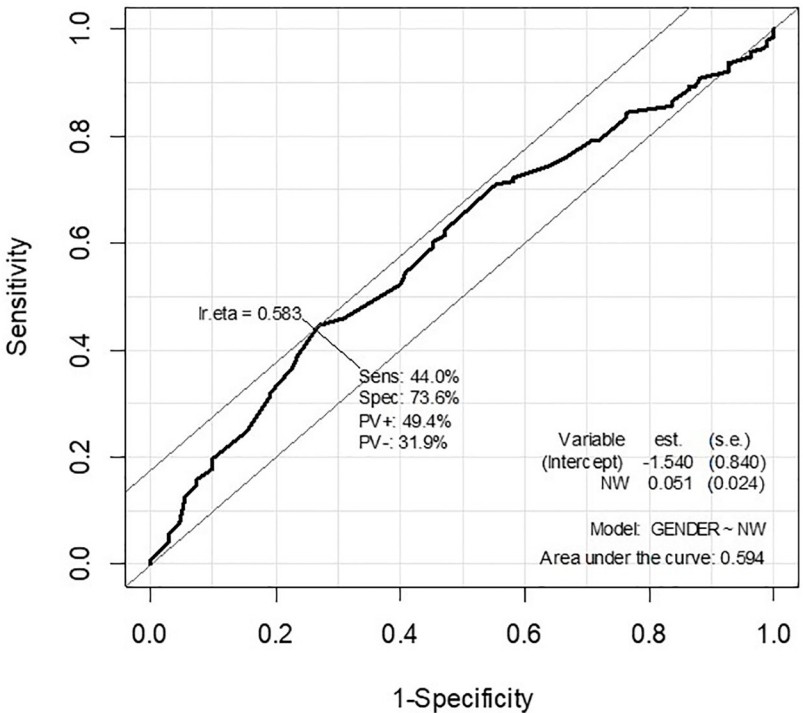

**Fig 2. Receiver operating characteristic curve for prediction analysis of sex from NW.**

The data were analyzed in R software (version 3.6.3, R Core Team, R Foundation for Statistical Computing, Vienna, Austria). Initially, the data were tested for normal distribution, verifying a normal distribution for all the quantitative variables (p>0.05; Kolmogorov-Smirnov normality test). Sequentially, the values of the variables OH, OW, NW, NH, OI, NI, and RONI were compared between male and female subjects (Levene's test and Student's t-test), and dependence of outcome (sex prediction) on each variable of interest was evaluated by logistic regression, using female sex as the reference group. The female gender was chosen as the reference category because the measures of interest were lower for females and leaving this category as a reference makes the interpretation of the odds ratio findings more understandable. Receiver operating characteristic curves (ROCs) (Figs 2 and 3) were generated for the NW and NH variables, these being the most accurate predictors of sex. The best cutoff points for each variable of interest were obtained from the equation "Ir.eta = $1/1+e^{-(\alpha + \beta X)}$": 36.76 for NW measurements, and 50.00 for NH measurements. For the two variables under study, values below the cutoff score were interpreted as indicating female sex, while values above the cutoff score were taken to indicate male sex. In the second phase of this study, these cutoff points were tested in an independent sample of 157 other individuals. A contingency table (confusion matrix) was generated, along with the respective predictive values of sensitivity and specificity (Se and Sp), positive and negative predictive values (PV+ and PV-), the positive and negative likelihood ratios (LR+ and LR-), and accuracy (Acc), including their respective confidence intervals 95% (CI95%). The significance level was 5% for all analyses.

## Results

Two hundred fifty-one individuals were evaluated at the initial test phase of the study, 110 females and 141 males, with a mean age of 37.41 (±20.56) and 33.87 (±19.58), respectively,

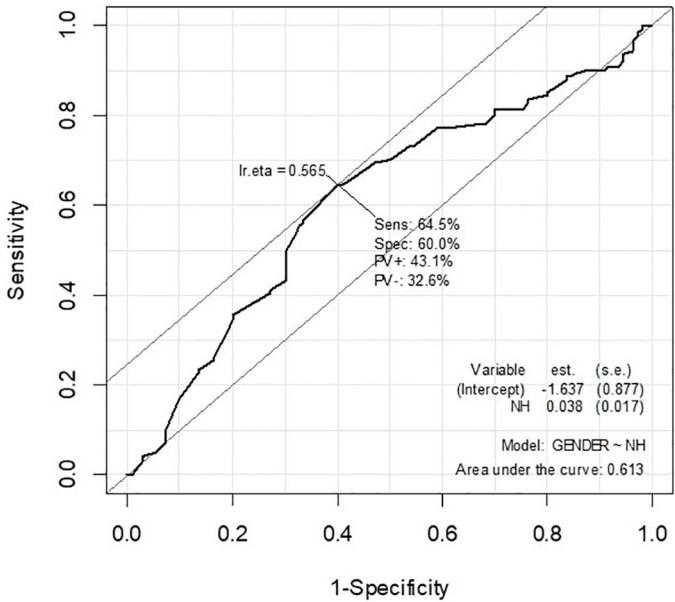

**Fig 3. Receiver operating characteristic curve for prediction analysis of sex from NH.**

there was no statistically significant difference in age between the sexes. (p = 0.232; Student's t-test).

Table 1 presents the distributions and comparisons of measurements for males and females.

Table 2 shows the results of the analysis for each predictor variable. It was not possible to fit a multiple model, with more than one variable being a predictor together with another, among the variables evaluated. For this reason, the univariate analysis is shown in Table 2, where it is observed that, in isolation, the variables NW and NH were configured as predictors of sex. For this reason, Table 2 describes only the univariate analysis.

In the data validation step (n = 157), the accuracy of sex prediction was 52.86% for NW and 64.96% for NH, considering the test cutoff points indicated (Table 3).

Table 3 describes the values obtained for the sample used in the test phase of the new cutoff points, which were applied to the test sample, generating the values of sensitivity (Se), specificity (Sp), positive predictive value (PV+), negative predictive value (PV-), positive likelihood (LR+), negative likelihood (LR-), and accuracy (Acc).

For NW and NH, moderate accuracies were obtained. Positive and negative predictive values were moderate and high, respectively, indicating moderate and high probabilities of correct classification, for females and males, respectively.

Table 3 also shows the positive and negative likelihood values (LR+ and LR-) of the NH and NW variables, that is the probability that subjects classified by the model as male truly are male, and the probability that subjects classified by the model as female truly are female.

## Discussion

Human beings are unique in terms of their physical growth and development, which means that related measurements can be used to identify individuals based on physical variability. The morphological features of the skull are useful in the identification of dead and living persons [26,27]. Measurements of facial anatomy, including sinuses, nasal cavity, or orbital cavity,

**Table 1. Distribution of measurements of OH, OW, NW, NH, OI, NI, and RONI, and comparison between males and females.**

| Variable | Predict categories | Measurement value Mean (SD) | p-value |
|---|---|---|---|
| OH | Female | 38.42 (±5.14) | 0.129 |
| | Male | 39.44 (±5.32) | |
| OW | Female | 41.54 (±4.01) | 0.252 |
| | Male | 42.15 (±4.36) | |
| NW | Female | 34.15 (±5.07) | 0.030 |
| | Male | 35.66 (±5.70) | |
| NH | Female | 49.14 (±7.36) | 0.028 |
| | Male | 51.25 (±7.59) | |
| OI | Female | 0.92 (±0.09) | 0.363 |
| | Male | 0.93 (±0.10) | |
| NI | Female | 0.70 (±0.09) | 0.984 |
| | Male | 0.70 (±0.09) | |
| RONI | Female | 1.33 (±0.20) | 0.453 |
| | Male | 1.36 (±0.23) | |

OH (orbital height); OW (orbital width); NW (nasal width); NH (nasal height); OI (orbital index = OH/OW); NI (nasal index = NW/NH); RONI (OI/NI). Student's t-test. Significance level = 5%.

are now widely used in forensic identification, even if it is necessary to carefully differentiate between different ethnic groups. Face measurements are also made for various health purposes, such as in the field of dental prostheses, cosmetic surgery, orthodontics and face masks. In particular, face analysis and proportions have been shown to be important for facial plastic surgeons to evaluate the face during the planning stages of facial reconstructive and cosmetic surgery [28].

**Table 2. Results of sex prediction from OH, OW, NW, NH, OI, NI, and RONI.**

| Variable | Sex | p-value | OR (CI 95%) |
|---|---|---|---|
| OH | Female | 0.130 | 1 |
| | Male | | 1.03 (0.98–1.08) |
| OW | Female | 0.252 | 1 |
| | Male | | 1.03 (1.02–1.10) |
| NW | Female | 0.031 | 1 |
| | Male | | 1.05 (1.01–1.08) |
| NH | Female | 0.030 | 1 |
| | Male | | 1.03 (1.01–1.05) |
| OI | Female | 0.376 | 1 |
| | Male | | 3.12 (0.61–5.62) |
| NI | Female | 0.989 | 1 |
| | Male | | 0.98 (-1.56–3.53) |
| RONI | Female | 0.462 | 1 |
| | Male | | 1.53 (0.39–2.66) |

OH (orbital height); OW (orbital width); NW (nasal width); NH (nasal height); OI (orbital index = OH/OW); NI (nasal index = NW/NH); RONI (OI/NI). Logistic regression. OR = Odds ratio. IC 95% = Confidence interval 95%. Significance level = 5%.

Table 3. Contingency table and predictive values for classification of individuals based on NW and NH measurements.

| Variable | Real classification | | Se | Sp | PV+ | PV- | LR+ | LR- | Acc |
|---|---|---|---|---|---|---|---|---|---|
| | Female | Male | | | | | | | |
| NW classification | Female | 43 | 58 | 72.88% (65.93–79.83) | 40.81% (33.12–48.50) | 42.57% (34.84–50.30) | 71.42% (64.35–78.49) | 1.23 (0.54–1.92) | 0.66 (0.58–0.73) | 52.86% (45.05–60.67) |
| | Male | 16 | 40 | | | | | | | |
| NH classification | Female | 33 | 29 | 55.93% (48.16–63.70) | 70.40% (63.26–77.54) | 53.22% (45.41–61.03) | 72.63% (65.66–79.60) | 1.88 (1.05–2.71) | 0.62 (0.54–0.69) | 64.96% (57.50–72.42) |
| | Male | 26 | 69 | | | | | | | |

Se = Sensitivity; Sp = Specificity; PV+ = Positive predictive value; PV- = Negative predictive value; LR+ = Positive likelihood ratio; LR- = Negative likelihood ratio; Acc = Accuracy.

Craniometric parameters including NI and OI have been investigated in prior research to estimate sex in forensic medicine. Preliminary knowledge of these measurements is essential for correct application since they vary from one population to another. The literature shows that there are significant differences in orbital and nasal morphometry between individuals in relation to age, sex, and ethnicity [29,30].

This study was conducted to establish the orbital and nasal indexes of Kosovar adults using skull CT scans. Previous studies have shown that measurements for the dimensions of orbital and nasal cavities taken from human skulls are similar to those obtained by CT scans [31], and the consistency of measurements using CT images has been evaluated in the last decade [32]. Our analysis was performed on head CT scans of patients with headache disorders, neurological deficits (e.g., poststroke), epilepsy, and other medical issues. This research excluded individuals with a history of bone pathology, and the sample can therefore be considered a "normal population" [33].

The human nose, as the most protruding part of the face, is variable in size and shape [34,35]. Divergence of nose shape and prominence can be explained by ethnic influences and environmental and climatic conditions [34–36]. In the present study using CT scans, the measurements of both the nasal aperture and orbital cavity were higher in males compared to females. These observations agree with the results of Vidya et al., who examined dry skulls of South Indian origin [36], and with those of Nasir et al., who studied nasal indexes across four Indian states [37]. Similarly, the nasal index, calculated from height and weight measurements of the nasal cavity, was found to be higher in males than in females in this research, in line with the findings of Staka et al. (on another Kosovar population), Hegazy (on Egyptians), Sforza et al., and Franciscus and Long [21,22,38]. Our results also showed that Kosovar adult males have a mean orbital index of 0.93 (0.10), while the equivalent score for adult females is 0.92 (0.09) (Table 1). Results from this study concur with previous research that found a significant sex difference (p<0.05) in the orbital indexes of Indian and Nigerian adults [39]. The present study does not support other investigations that have noted a higher OI in females than in males. The differences seen in these populations may be attributed to the different age groups of subjects across various studies [40]. Also, in our study the mean orbital height and weight measurements for females and males are half of those recorded in Indian, Nigerian, Malawian, Bini, and Urhobo populations [41]. We found a significant difference in the orbital index between males and females, mainly because males exhibit wider orbital cavities than females. Some authors have reported differences in the volume of OI between males and females, whereas others have shown no such difference [42–45].

Lusted and Keats reported that orbital parameters obtained from radiographs are slightly different from those obtained from direct measurement of human skulls and that this difference could be attributed to the magnification factor of X-ray machines [46]. Indeed, the results

of the present study might have been affected by the magnification factor and therefore may not present completely true measurements of the orbital or nasal cavity. Other differences between our findings and those of other authors could be due to different patterns of craniofacial growth caused by racial, ethnic, social, environmental, and nutritional factors [46–48].

Differences in nose shape and appearance are influenced by ethnic factors and environmental climatic conditions. A study on the Dayak Kenyah population in North Kalimantan, Indonesia, found that the predominant nasal shape was mesorrhine and attributed this to environmental factors, concluding that differences between nasal shapes may be used to trace geographic origins [49]. In a study on a South Indian population, using MDCT 2D scans to measure nasal index and nasal parameters, Kotian et al. [25] found that the measurements of nasal aperture were greater in males compared to females; these findings are in agreement with the present study. Regardless of sexual dimorphism, it is predictable that ethnic groups in the same climatic regions should have similarities in their nasal indexes. In any case, the point that is most relevant for forensic investigations is that anthropometric data of the face and nose would be useful for sex determination, and Svitlana and Themozhi have suggested that nasal proportions are indeed a useful anthropometrical tool in determination of sex in forensic science and classification of fossil remains [50].

This study indicates that NW and NH have potential to classify individuals as female and male with predictive accuracy of around 52.86% and 64.96%, respectively. Anthropological studies agree that variations in the nasal index are related to the climate. Variation in the bony nasal cavity probably stems from man's ability to adapt to the environment. The peculiarity of this study is the fact that a different predictivity in the estimation of sex, with statistical significance, lies not in the calculation of the NI, but in the individual measurements which, if compared, form the NI (NW and NH). In particular, females tend to have a lower NW than that of males, as well as a higher NH. However, it should be noted that, as limitation, the study has moderate accuracy with not very high power to classify sex. In a larger pool of samples or in other Caucasian populations we could find even higher levels of accuracy.

In confirmation of the hypothesis that initiated this research, the statistically significant differences found between the two sexes as regards the nasal parameters could be traced back to a different process of bone growth of parts of the facial mass in males and females, due both to genetic factors typical of a specific population group and/or evolutionary-environmental factors established over time.

To understand whether this nasal anatomical variation is due to the adaptation of the Kosovar population to the specific environment or whether it concerns other ethnic groups, broadly speaking, it would be useful to extend the analysis to other neighboring and non-neighboring populations. Furthermore, for the purpose of forensic identification, it will be useful to verify any anthropometric differences in relation to age, as well as to supplement the method with other already known or innovative methods for age estimation [51,52].

## Conclusion

Sex estimation is essential for completing an individual's biological profile and therefore constitutes an important part of the human identification process in the forensic field.

The current anthropometric study found that the metric parameters necessary for the calculation of NI showed pronounced sexual dimorphism in a large Kosovar population. Since the equation developed in the stepwise discriminant analysis was intended for this specific population, it may produce different results for other population groups. Further studies are recommended to evaluate and characterize orbital and nasal parameters among different populations; it could be useful test the discriminant function in populations from different

countries, and study the dependence of anthropometric data on sex and age. Therefore, conducting a comparative study between different groups would be very interesting in the field of personal identification for forensic purposes. In the future, studies could be conducted to compare the measurements made on CT scans with those performed directly on the skulls and verify if there is an anthropometric variability that could make the results different.

## Supporting information

**S1 Data.**
(XLSX)

## Acknowledgments

Thanks to Jemma Dunnill for proofreading the manuscript.

## Author Contributions

**Conceptualization:** Roberto Scendoni, Roberto Cameriere.

**Data curation:** Jeta Kelmendi, Isabella Lima Arrais Ribeiro, Francesco De Micco.

**Formal analysis:** Francesco De Micco.

**Investigation:** Roberto Scendoni, Isabella Lima Arrais Ribeiro.

**Methodology:** Mariano Cingolani, Roberto Cameriere.

**Resources:** Jeta Kelmendi.

**Software:** Francesco De Micco.

**Supervision:** Mariano Cingolani.

**Validation:** Roberto Cameriere.

**Visualization:** Jeta Kelmendi, Mariano Cingolani.

**Writing – original draft:** Roberto Scendoni, Isabella Lima Arrais Ribeiro.

**Writing – review & editing:** Roberto Cameriere.

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
