## [Decision Letter · Decision Letter 0]

11 Aug 2022

PONE-D-22-09865

Anthropometric analysis of orbital and nasal parameters for sexual dimorphism: new anatomical evidences in the field of personal identification

PLOS ONE

Dear Dr. De Micco,

Thank you for submitting your manuscript to PLOS ONE. After careful consideration, we feel that it has merit but does not fully meet PLOS ONE’s publication criteria as it currently stands. Therefore, we invite you to submit a revised version of the manuscript that addresses the points raised during the review process.

We look forward to receiving your revised manuscript.

Kind regards,

Kapil Amgain

Academic Editor

PLOS ONE

https://journals.plos.org/plosone/s/file?id=ba62/PLOSOne_formatting_sample_title_authors_affiliations.pdf".

Additional Editor Comments:

Dear Author,

The manuscript covers the very important issue of Anthropometric of orbital and nasal parameters which is useful for sexual dimorphism and it is helpful in the field of personal identification. However you need to revise some aspect of the manuscript. Please find the attached comments and revise it accordingly.

Thank you

Reviewers' comments:

Reviewer's Responses to Questions

**Comments to the Author**

1. Is the manuscript technically sound, and do the data support the conclusions?

Reviewer #1: Yes

Reviewer #2: Yes

2. Has the statistical analysis been performed appropriately and rigorously? 

Reviewer #1: Yes

Reviewer #2: Yes

3. Have the authors made all data underlying the findings in their manuscript fully available?

Reviewer #1: Yes

Reviewer #2: Yes

4. Is the manuscript presented in an intelligible fashion and written in standard English?

Reviewer #1: Yes

Reviewer #2: Yes

5. Review Comments to the Author

Reviewer #1: Please adhere to the comment. The references have to be cited properly. The Vancouver style has not been used properly in few of the articles. Please maintain it. The liitation of the study has to be mentioned too.

Reviewer #2: 114: Can you make clear why you choose female sex as a reference group?

135: The authors did not establish a correlation between sex difference and mean age why?

182: The authors did not establish a correlation between sex difference and mean age why?

221: Why not to compare the finding of CT scan with direct measurement of some subjects to verify whether the measurement is affected by the magnification factor or not?

6. PLOS authors have the option to publish the peer review history of their article (what does this mean?). If published, this will include your full peer review and any attached files.

Reviewer #1: No

Reviewer #2: No

---

## [Author Response · Author response to Decision Letter 0]

2 Sep 2022

Dear Academic Editor

Prof. Kapil Amgain

thank you very much for your kind mail and the precious reviews included in your response. 

We are glad to inform You that all the criticisms raised by Reviewers have been accepted.

The manuscript has been largely revised and improved according to these criticisms.

In this regard, we really thank the Reviewers for the relevant improvement to the article

This is our reply point-by-point.

Reviewer #1

a) Reply to issue no. 1: “Dear author the article you have cited here is speaking of only orbital index and that too not about evoution. please revise and be specific while citing articles”

Thank you for your remark. We have added the following references:

- Oladipo GS, Olabiyi AO, Oremosu AA, NoronhaCC. Nasal indices among major ethnic groups in Southern Nigeria. SRE. 2007; 2(1): 20-22. 

- Ravichandran S, Yuvraj Babu K, Mohanraj KG. Correlation of facial and nasal index in gender determination. Drug Invention Today. 2018; 10(12).

b) Reply to issue no. 2: “What were the inclusion and exclusion criteria of the study?”

In the “Materials and Methods” section, we have added the following explanation: “Inclusion criteria were Kosovar males and females over the age of 18, who underwent CT scan to investigate neurological symptoms. Exclusion criteria included growth diseases, endocrine disorders or osteodystrophy, previous fractures of the of the skull and the facial massif. Finally, all blurred TC images were disregarded”

c) Reply to issue no. 3: “How was this sample size calculated?”

In the “Materials and Methods” section, we have added the following explanation: “The sample of the present study was calculated based on the effects observed in the study by Kotian et al. From this study, the mean effect size of difference between groups of 0.20 (Cohen's d) was observed. Adopting a 5% type I error (95% study confidence level), a 20% type II error (80% study power), the effect size observed in the mentioned study and, considering the comparison tests of the groups of the bilateral type (two-sided), it was calculated that at least 394 sample units were necessary for the accomplishment of the study. As they are available, we conducted the study with 408 sample units. The calculations were performed using the G*Power statistical software”

d) Reply to issue no. 4: “Dear author, is this line so important for the result section.”

Thanks for the suggestion. We have deleted the previous sentence and explained our thinking better: “It was not possible to fit a multiple model, with more than one variable being a predictor together with another, among the variables evaluated. For this reason, the univariate analysis is shown in Table 2, where it is observed that, in isolation, the variables NW and NH were configured as predictors of sex”

e) Reply to issue no. 5: “This is not a correct statement over here, the increase in the number of samples depends upon the sample size calculation and in this study the authors have enough sample size. however, the basis of sample taken has not been shown so the methodology clearly needs how the sample size was obtained and then the revision of the line 263 has to be thought of”

The following sentence has been deleted “to increase the number of samples”. The answer to criticism No. 3 shows how the sample was calculated.

f) Reply to issues nos. 6, 7, 8, 9 and 10.

The references have been corrected

***

Reviewer #2

a) Reply to issue no. 1: “Can you make clear why you choose female sex as a reference group?”

We have added the following sentence to better explain the reason for the choice: “The female gender was chosen as the reference category because the measures of interest were lower for females and leaving this category as a reference makes the interpretation of the odds ratio findings more understandable”

b) Reply to issues nos. 2 and 3: “The authors did not establish a correlation between sex difference and mean age why?” and “The authors did not establish a correlation between sex difference and mean age why?”

Thanks to the reviewer for the remark. The comparison was made and reported in the 'Results' section. Below is the sentence: “251 individuals were evaluated, 110 females and 141 males, with a mean age of 37.41 (±20.56) and 33.87 (±19.58), respectively, there was no statistically significant difference in age between the sexes. (p=0.232; Student’s t-test)”. 

c) Reply to issue no. 4: “Besides forensic identification, please add some other reasons why facial measurements are necessary”

In the "Discussion" section, we have added the following sentence (with reference) to better explain some other reasons why facial measurements are necessary: “Face measurements are also made for various health purposes, such as in the field of dental prostheses, cosmetic surgery, orthodontics and face masks. In particular, face analysis and proportions have been shown to be important for facial plastic surgeons to evaluate the face during the planning stages of facial reconstructive and cosmetic surgery 

- Shahbazi Z, Ardalani H, Maleki M. Aesthetics of Numerical Proportions in Human Cosmetic Surgery. World J Plast Surg. 2019 Jan;8(1):78-84. 

d) Reply to issue no. 5: “Why not to compare the finding of CT scan with direct measurement of some subjects to verify whether the measurement is affected by the magnification factor or not?”

Good observation, this study is in anticipation. In the conclusions section we added this sentence: “In the future, studies could be conducted to compare the measurements made on CT scans with those performed directly on the skulls and verify if there is an anthropometric variability that could make the results different”

For details we invite You to check the file revised and updated in attachment. 

We really hope that the comments raised by the Reviewers have been satisfied in this manner. 

We look forward to hearing from You.

Sincerely,

Dr. Francesco De Micco, M.D., Ph.D

Corresponding author

---

## [Editor Report · Decision Letter 1]

15 Sep 2022

PONE-D-22-09865R1Anthropometric analysis of orbital and nasal parameters for sexual dimorphism: new anatomical evidences in the field of personal identificationPLOS ONE

Dear Dr. De Micco,

Thank you for submitting your manuscript to PLOS ONE. After careful consideration, we feel that it has merit but does not fully meet PLOS ONE’s publication criteria as it currently stands. Therefore, we invite you to submit a revised version of the manuscript that addresses the points raised during the review process.

We look forward to receiving your revised manuscript.

Kind regards,

Kapil Amgain

Academic Editor

PLOS ONE

Journal Requirements:

Additional Editor Comments (if provided):

Dear Author,

The manuscript cover the very important topic of sexual dimorphism. Revise the manuscript as per the feedback provided to proceed further process.

Thank you
---

## [Author Response · Author response to Decision Letter 1]

6 Jan 2023

Dear Editor in Chief

thank you very much for your kind mail and the precious reviews included in your response. 

We are glad to inform You that all the criticisms raised by Reviewers have been accepted.

The manuscript has been largely revised and improved according to these criticisms.

In this regard, we really thank the Reviewers for the relevant improvement to the article

This is our reply point-by-point.

Reviewer #1

a) Reply to issue no. 1: “Dear author the article you have cited here is speaking of only orbital index and that too not about evoution. please revise and be specific while citing articles”

Thank you for your remark. We have added the following references:

- Oladipo GS, Olabiyi AO, Oremosu AA, NoronhaCC. Nasal indices among major ethnic groups in Southern Nigeria. SRE. 2007; 2(1): 20-22. 

- Ravichandran S, Yuvraj Babu K, Mohanraj KG. Correlation of facial and nasal index in gender determination. Drug Invention Today. 2018; 10(12).

b) Reply to issue no. 2: “What were the inclusion and exclusion criteria of the study?”

In the “Materials and Methods” section, we have added the following explanation: “Inclusion criteria were Kosovar males and females over the age of 18, who underwent CT scan to investigate neurological symptoms. Exclusion criteria included growth diseases, endocrine disorders or osteodystrophy, previous fractures of the of the skull and the facial massif. Finally, all blurred TC images were disregarded”

c) Reply to issue no. 3: “How was this sample size calculated?”

In the “Materials and Methods” section, we have added the following explanation: “The sample of the present study was calculated based on the effects observed in the study by Kotian et al. From this study, the mean effect size of difference between groups of 0.20 (Cohen's d) was observed. Adopting a 5% type I error (95% study confidence level), a 20% type II error (80% study power), the effect size observed in the mentioned study and, considering the comparison tests of the groups of the bilateral type (two-sided), it was calculated that at least 394 sample units were necessary for the accomplishment of the study. As they are available, we conducted the study with 408 sample units. The calculations were performed using the G*Power statistical software”

d) Reply to issue no. 4: “Dear author, is this line so important for the result section.”

Thanks for the suggestion. We have deleted the previous sentence and explained our thinking better: “It was not possible to fit a multiple model, with more than one variable being a predictor together with another, among the variables evaluated. For this reason, the univariate analysis is shown in Table 2, where it is observed that, in isolation, the variables NW and NH were configured as predictors of sex”

e) Reply to issue no. 5: “This is not a correct statement over here, the increase in the number of samples depends upon the sample size calculation and in this study the authors have enough sample size. however, the basis of sample taken has not been shown so the methodology clearly needs how the sample size was obtained and then the revision of the line 263 has to be thought of”

The following sentence has been deleted “to increase the number of samples”. The answer to criticism No. 3 shows how the sample was calculated.

f) Reply to issues nos. 6, 7, 8, 9 and 10.

The references have been corrected

***

Reviewer #2

a) Reply to issue no. 1: “Can you make clear why you choose female sex as a reference group?”

We have added the following sentence to better explain the reason for the choice: “The female gender was chosen as the reference category because the measures of interest were lower for females and leaving this category as a reference makes the interpretation of the odds ratio findings more understandable”

b) Reply to issues nos. 2 and 3: “The authors did not establish a correlation between sex difference and mean age why?” and “The authors did not establish a correlation between sex difference and mean age why?”

Thanks to the reviewer for the remark. The comparison was made and reported in the 'Results' section. Below is the sentence: “251 individuals were evaluated, 110 females and 141 males, with a mean age of 37.41 (±20.56) and 33.87 (±19.58), respectively, there was no statistically significant difference in age between the sexes. (p=0.232; Student’s t-test)”. 

c) Reply to issue no. 4: “Besides forensic identification, please add some other reasons why facial measurements are necessary”

In the "Discussion" section, we have added the following sentence (with reference) to better explain some other reasons why facial measurements are necessary: “Face measurements are also made for various health purposes, such as in the field of dental prostheses, cosmetic surgery, orthodontics and face masks. In particular, face analysis and proportions have been shown to be important for facial plastic surgeons to evaluate the face during the planning stages of facial reconstructive and cosmetic surgery 

- Shahbazi Z, Ardalani H, Maleki M. Aesthetics of Numerical Proportions in Human Cosmetic Surgery. World J Plast Surg. 2019 Jan;8(1):78-84. 

d) Reply to issue no. 5: “Why not to compare the finding of CT scan with direct measurement of some subjects to verify whether the measurement is affected by the magnification factor or not?”

Good observation, this study is in anticipation. In the conclusions section we added this sentence: “In the future, studies could be conducted to compare the measurements made on CT scans with those performed directly on the skulls and verify if there is an anthropometric variability that could make the results different”

For details we invite You to check the file revised and updated in attachment. 

We really hope that the comments raised by the Reviewers have been satisfied in this manner. 

We look forward to hearing from You.

Sincerely,

Dr. Francesco De Micco, M.D., Ph.D

Corresponding author

---

## [Decision Letter · Decision Letter 2]

6 Mar 2023

PONE-D-22-09865R2Anthropometric analysis of orbital and nasal parameters for sexual dimorphism: new anatomical evidences in the field of personal identificationPLOS ONE

Dear Dr. De Micco,

Thank you for submitting your manuscript to PLOS ONE. After careful consideration, we feel that it has merit but does not fully meet PLOS ONE’s publication criteria as it currently stands. Therefore, we invite you to submit a revised version of the manuscript that addresses the points raised during the review process.

We look forward to receiving your revised manuscript.

Kind regards,

Johari Yap Abdullah, B.S. & I.T, GradDip ICT, M.Sc, Ph.D.

Academic Editor

PLOS ONE

Journal Requirements:

Reviewers' comments:

Reviewer's Responses to Questions

**Comments to the Author**

1. If the authors have adequately addressed your comments raised in a previous round of review and you feel that this manuscript is now acceptable for publication, you may indicate that here to bypass the “Comments to the Author” section, enter your conflict of interest statement in the “Confidential to Editor” section, and submit your "Accept" recommendation.

Reviewer #1: All comments have been addressed

Reviewer #2: (No Response)

Reviewer #3: (No Response)

Reviewer #4: (No Response)

2. Is the manuscript technically sound, and do the data support the conclusions?

Reviewer #1: Yes

Reviewer #2: (No Response)

Reviewer #3: Yes

Reviewer #4: Partly

3. Has the statistical analysis been performed appropriately and rigorously? 

Reviewer #1: Yes

Reviewer #2: (No Response)

Reviewer #3: Yes

Reviewer #4: Yes

4. Have the authors made all data underlying the findings in their manuscript fully available?

Reviewer #1: Yes

Reviewer #2: (No Response)

Reviewer #3: No

Reviewer #4: Yes

5. Is the manuscript presented in an intelligible fashion and written in standard English?

Reviewer #1: Yes

Reviewer #2: (No Response)

Reviewer #3: Yes

Reviewer #4: Yes

6. Review Comments to the Author

Reviewer #1: On line no 146, please do not start your result section with numeral (251) rather it should be Two hundred fifty one.

Reviewer #2: (No Response)

Reviewer #3: Reviewer comments:

1. General response to the research:

a. This study aimed to ascertain whether there are sex differences in the orbital and/or nasal indexes and/or the single measurements used to calculate these variables in a Kosovar population.

Craniometric parameters including NI and OI have been investigated in previous studies in forensic medicine. However, the literature shows that there are significant differences in these parameters between individuals in relation to age, sex, and ethnicity. Although, this study does not show any novel techniques and methods, it did study a specific population (Different ethnicity) which is the Kosovar population.

b. The methodology in this study is clear with some suggestions to be addressed by the authors. The overall Results and discussion are good. English language used is very good. There are some problems with the references in the references section.

c. The supplementary data or findings were not provided. No link or deposition to a public repository was found in the submitted manuscript, although the authors mentioned “Yes - all data are fully available without restriction”

d. It is better to follow the STROBE guidelines in reporting data for retrospective studies.

e. This article, in my opinion, suits this journal’s aim and vision.

f. All the comments were highlighted in the pdf starting from page 44.

2. Title: Please mention the type of study at the end of the title.

3. Abstract:

a. Please re-write the abstract according to the journal guidelines.

b. Kindly delete the abbreviations that are not useful.

c. Please write CT in full.

4. Introduction:

a. No hypothesis was presented in this section.

b. Please add reference for the statement that end in row 65.

c. Please cite the studies that you have mentioned (The sentence in row 66 and 67)

5. Methodology: The authors did not follow completely STROBE guidelines in reporting their methodology.

a. Describe the setting, locations, and relevant dates of selecting and measuring the data.

b. State the Ethical approval in the beginning of this section. This study involving medical records. so Ethical approval by a review board is needed.

c. Please add the ethnic group in your inclusion criteria.

d. Correct the mistake in row 86

e. In row 93, please size to the sample.

f. State any risk of potential bias if necessary (the difference in the sample size of the initial test phase between males (n=141) and females (n=110)).

g. How did you manage any possible risk of the X-ray machine magnification? Was there any criteria when selecting a proper x-ray to be studied?

h. In figure 1, please state the dimensions of the film and indicate which side is the left or right.

6. Results:

a. Please add this in the sentence of row 146. " 251 individuals were evaluated at the initial test phase of the study"

b. In row 164, please change predictive to likelihood

7. Discussion: Please add a discussion on your hypothesis depending on what you did write in the introduction section.

8. Conclusion: Please remove the 40-42 or add them into the discussion. It is advisable not to include any new references here as the conclusion is withdrawn from your data and discussion.

9. References: Few mistakes were found such as page number, journal name, etc. Kindly check the journal guidelines again before you do any corrections. Some mistakes were highlighted in the pdf. Kindly check all the references and correct whenever needed.

10. Figures: In figure 1, please state the dimensions of the film and indicate which side is the left or right.

Reviewer #4: Dear authors,

Thank you for the paper. I did enjoy reading it. One issue which is worth investigating more is the age differences for the population. Even if this paper is looking into sex differences, age is important for two reasons.

A. Bone remodelling as a person age normally.

B. Edentulous causing the maxilla and mandible bone to remodel differently.

1. Reviewer #2 has asked regarding age and a response has been given though the range is quite big. Is it possible for the authors to do analysis on a 5 year or 10 year age range of individuals disregarding the sex? Example 18-27 years old, 28-37 ..... There should be a difference in the bone morphology of elderly adults compared to 18 year old healthy individuals. Based on the raw data obtained, there should be enough to just

2. The paper mentions exclusion of pathological deformed bones. Does edentulous fall under this category? The maxilla bone can appear normal, but the person may not have maxilla teeth.

Please refer to this paper for why the orbit is affected by aging:

Pessa, J.E., 2000. An algorithm of facial aging: verification of Lambros’s theory by three-dimensional stereolithography, with reference to the pathogenesis of midfacial aging, scleral show, and the lateral suborbital trough deformity. Plastic and reconstructive surgery, 106(2), pp.479-488.

7. PLOS authors have the option to publish the peer review history of their article (what does this mean?). If published, this will include your full peer review and any attached files.

Reviewer #1: No

Reviewer #2: No

Reviewer #3: No

Reviewer #4: No

---

## [Author Response · Author response to Decision Letter 2]

10 Mar 2023

RESPONSE TO REVIEWERS

Dear Academic Editor

thank you very much for your kind mail and the precious reviews included in your response. 

We are glad to inform You that all the criticisms raised by Reviewers have been accepted.

The manuscript has been revised and improved according to these criticisms.

In this regard, we really thank the Reviewers for the relevant improvement to the article

This is our reply point-by-point.

Reviewer #1

Dear Reviewer #1,

all authors are grateful to you for considering our manuscript interesting and for your suggestions.

We accepted your proposals and implemented our manuscript.

a) Reply to issue: “On line no 146, please do not start your result section with numeral (251) rather it should be Two hundred fifty one”.

Thank you for your remark. Taking up the suggestion, we have changed from numbers to letters

Reviewer #3

Dear Reviewer #3,

all authors are grateful to you for considering our manuscript interesting and for your suggestions.

We accepted your proposals and implemented our manuscript.

a) Reply to issue no. 1c: “The supplementary data or findings were not provided. No link or deposition to a public repository was found in the submitted manuscript, although the authors mentioned “Yes - all data are fully available without restriction”

We have uploaded an excel file as supplementary data

b) Reply to issue no. 1d: “It is better to follow the STROBE guidelines in reporting data for retrospective studies”.

We have modified according to STROBE guidelines

c) Reply to issue no. 2: “Please mention the type of study at the end of the title”.

We have added “through a retrospective observational study” in the title

d) Reply to issue no. 3: “Abstract: a. Please re-write the abstract according to the journal guidelines. b. Kindly delete the abbreviations that are not useful. c. Please write CT in full”

The abstract was rewritten with the aim of (i) describe the main objective of the study; (i) explain how the study was done, including any model organisms used, without methodological detail; (i) summarize the most important results and their significance; (i) not exceed 300 words. All this according to Plosone guidelines.

e) Reply to issue no. 4a: “No hypothesis was presented in this section”

We have stated our research hypothesis by adding the following sentence: “The authors assume that there may be differences between the two sexes at least in one or more parameters necessary to calculate either the nasal index or the orthotic index in the population under examination. If the hypothesis is confirmed, on the basis of the results obtained, the authors will provide a plausible explanation in this regard”.

f) Reply to issue no. 4b: “Please add reference for the statement that end in row 65.”

The following reference is made to the sentence: Cameron N, Lones LL. Growth, maturation and age, in: Black S, Aggrawal A, Payne-James J, editors. Age estimation in the living. The practitioners guide. Chichester, West Sussex, UK; 2010. pp. 95-120.

g) Reply to issue no. 4c: “Please cite the studies that you have mentioned (The sentence in row 66 and 67)”

The following references are made to the sentence: 

- Botwe BO, Sule DS, Ismael AM. Radiologic evaluation of orbital index among Ghanaians using CT scan. J Physiol Anthropol. 2017;36:29.

- Gupta V, Prabhakar A, Yadav M, Khandelwal N. Computed tomography imaging-based normative orbital measurement in Indian population. Indian J Ophthalmol. 2019; 67:659–63.

- Iscan M. Forensic anthropology of sex and body size. Forensic Sci Int. 2005; 147:107–12

- Russel SM, Frank-Ito DO. Gender Differences in Nasal Anatomy and Function Among Caucasians. Facial Plast Surg Aesthet Med. 2023; 2:145-152

h) Reply to issue no. 5a: “Describe the setting, locations, and relevant dates of selecting and measuring the data”.

We reworked the sentence: “The CT scans were anonymously extracted and limited data were extracted that were strictly necessary and relevant for the conduct of the study. The extracted data are all available in the manuscript. CT examinations were analyzed from 1 February to 15 March 2022”

i) Reply to issue no. 5b: “State the Ethical approval in the beginning of this section. This study involving medical records. so Ethical approval by a review board is needed”

We reworked the sentence: “The study was approved by the Ethical Issues Committee of the Kosovo Chamber of Dentists (approval no. 27 of 18 May 2022)”. 

The Ethics Committee's approval document was submitted

j) Reply to issue no. 5c: “Please add the ethnic group in your inclusion criteria”

We reworked the sentence: “The inclusion criteria were males and females of Kosovar ethnicity …”

k) Reply to issue no. 5d: “Correct the mistake in row 86”

TC has been corrected in CT

l) Reply to issue no. 5e: “In row 93, please size to the sample”

Addition done

m) Reply to issue no. 5f: “State any risk of potential bias if necessary (the difference in the sample size of the initial test phase between males (n=141) and females (n=110))”

We added the sentence: “The random splitting of the sub-samples could represent a potential research bias”

n) Reply to issue no. 6f: “How did you manage any possible risk of the X-ray machine magnification? Was there any criteria when selecting a proper x-ray to be studied?”

Regarding the possibility of magnification, the CTs were performed in standard projections. Therefore, we have a reasonable certainty that no interpretative alterations are possible. In addition, the software used guarantees perfect measurement in millimetres. 

This certainty is also the result of the exclusion criteria. And here, we answer the second question: all CTs of patients with growth disorders, endocrine disorders or osteodystrophy, previous fractures of the skull and facial massif were excluded.

o) Reply to issue no. 5h: “In figure 1, please state the dimensions of the film and indicate which side is the left or right”.

With regard to size, only measurements of interest were taken for the parameters required for the study (orbital and nasal). As for the right and left side, these are common CT images in antero-posterior projection.

p) Reply to issue no. 6a: “Please add this in the sentence of row 146. "251 individuals were evaluated at the initial test phase of the study".

We reworked the sentence: “Two hundred fifty-one individuals were evaluated at the initial test phase of the study”

q) Reply to issue no. 6b: “In row 164, please change predictive to likelihood”

We reworked the sentence: “Table 3 also shows the positive and negative likelihood values (LR+ and LR-) …”

r) Reply to issue no. 7: “Discussion: Please add a discussion on your hypothesis depending on what you did write in the introduction section”

We have added the following sentence: “In confirmation of the hypothesis that initiated this research, the statistically significant differences found between the two sexes as regards the nasal parameters could be traced back to a different process of bone growth of parts of the facial mass in males and females, due both to genetic factors typical of a specific population group and/or evolutionary-environmental factors established over time”.

s) Reply to issue no. 8: “Please remove the 40-42 or add them into the discussion. It is advisable not to include any new references here as the conclusion is withdrawn from your data and discussion”.

Indeed, these references are more suitable for introduction.

t) Reply to issue no. 9: “Few mistakes were found such as page number, journal name, etc. Kindly check the journal guidelines again before you do any corrections. Some mistakes were highlighted in the pdf. Kindly check all the references and correct whenever needed”.

We revised the layout of all references

u) Reply to issue no. 10: “Figures: In figure 1, please state the dimensions of the film and indicate which side is the left or right”

Please kindly compare with the answer to point “n”,

Reviewer #4

Dear Reviewer #4,

all authors are grateful to you for considering our manuscript interesting and for your suggestions.

We accepted your proposals and implemented our manuscript.

a) Reply to issue no. 1: “Reviewer #2 has asked regarding age and a response has been given though the range is quite big. Is it possible for the authors to do analysis on a 5 year or 10 year age range of individuals disregarding the sex? Example 18-27 years old, 28-37 ..... There should be a difference in the bone morphology of elderly adults compared to 18 year old healthy individuals. Based on the raw data obtained, there should be enough to just”.

Thank you for your remark. The observation is very pertinent. However, as it stands, the study has the limitations mentioned at the end of the paper. Specifically, the study showed moderate accuracy with not very high sex classification power. We should understand in the future whether this nasal anatomical variation is due to the adaptation of the Kosovar population to the specific environment or whether it affects other ethnic groups, in general.

Therefore, we also enthusiastically welcome and will evaluate in future studies the aspect suggested by reviewer #4 to see if and how bone variability might impact the method.

For this reason, we have added the following sentence: “Furthermore, for the purpose of forensic identification, it will be useful to verify any anthropometric differences in relation to age, as well as to supplement the method with other already known or innovative methods for age estimation”

b) Reply to issue no. 2: “The paper mentions exclusion of pathological deformed bones. Does edentulous fall under this category? The maxilla bone can appear normal, but the person may not have maxilla teeth. Please refer to this paper for why the orbit is affected by aging: Pessa, J.E., 2000. An algorithm of facial aging: verification of Lambros’s theory by three-dimensional stereolithography, with reference to the pathogenesis of midfacial aging, scleral show, and the lateral suborbital trough deformity. Plastic and reconstructive surgery, 106(2), pp.479-488”

We had no cases of edentualia. However, they would not have been excluded. The exclusion criteria were made explicit in the study and concern: growth diseases, endocrine disorders or osteodystrophy, previous fractures of the of the skull and the facial massif.

We gladly read the suggested paper (Pessa JE, 2000) and included it to make our discussion on age-related morphological variability even more 'evidence-based'.

For details we invite You to check the file revised and updated in attachment. 

We really hope that the comments raised by the Reviewers have been satisfied in this manner. 

We look forward to hearing from You.

Sincerely,

Dr. Francesco De Micco, M.D., Ph.D

Corresponding author

---

## [Decision Letter · Decision Letter 3]

27 Mar 2023

Anthropometric analysis of orbital and nasal parameters for sexual dimorphism: new anatomical evidences in the field of personal identification through a retrospective observational study

PONE-D-22-09865R3

Dear Dr. De Micco,

We’re pleased to inform you that your manuscript has been judged scientifically suitable for publication and will be formally accepted for publication once it meets all outstanding technical requirements.

Kind regards,

Johari Yap Abdullah, B.S. & I.T, GradDip ICT, M.Sc, Ph.D.

Academic Editor

PLOS ONE

Additional Editor Comments (optional):

Please address comments by Reviewer 2:

Proper citation method should be applied. I found the lot of errors in the reference section even after so many revisions given to the authors. This could be a serious issue to reject your paper.

Reviewers' comments:

Reviewer's Responses to Questions

**Comments to the Author**

1. If the authors have adequately addressed your comments raised in a previous round of review and you feel that this manuscript is now acceptable for publication, you may indicate that here to bypass the “Comments to the Author” section, enter your conflict of interest statement in the “Confidential to Editor” section, and submit your "Accept" recommendation.

Reviewer #1: All comments have been addressed

Reviewer #2: (No Response)

Reviewer #3: All comments have been addressed

Reviewer #4: All comments have been addressed

2. Is the manuscript technically sound, and do the data support the conclusions?

Reviewer #1: Yes

Reviewer #2: (No Response)

Reviewer #3: Yes

Reviewer #4: Yes

3. Has the statistical analysis been performed appropriately and rigorously? 

Reviewer #1: Yes

Reviewer #2: (No Response)

Reviewer #3: Yes

Reviewer #4: Yes

4. Have the authors made all data underlying the findings in their manuscript fully available?

Reviewer #1: Yes

Reviewer #2: (No Response)

Reviewer #3: Yes

Reviewer #4: Yes

5. Is the manuscript presented in an intelligible fashion and written in standard English?

Reviewer #1: Yes

Reviewer #2: (No Response)

Reviewer #3: Yes

Reviewer #4: Yes

6. Review Comments to the Author

Reviewer #1: (No Response)

Reviewer #2: Proper citation method should be applied. I found the lot of errors in the reference section even after so many revisions given to the authors. This could be a serious issue to reject your paper.

Reviewer #3: Dear Main/Corresponding author, thank you for addressing most of the points. No further comments from my side.

Reviewer #4: Dear authors,

Thank you for addressing the comments. The article can currently be accepted. It would be useful if another future study could focus on edentulous patients to see if it falls within the current research parameters.

7. PLOS authors have the option to publish the peer review history of their article (what does this mean?). If published, this will include your full peer review and any attached files.

Reviewer #1: No

Reviewer #2: No

Reviewer #3: No

Reviewer #4: No

---

## [Editor Report · Acceptance letter]

24 Apr 2023

PONE-D-22-09865R3 

Anthropometric analysis of orbital and nasal parameters for sexual dimorphism: new anatomical evidences in the field of personal identification through a retrospective observational study 

Dear Dr. De Micco:

I'm pleased to inform you that your manuscript has been deemed suitable for publication in PLOS ONE. Congratulations! Your manuscript is now with our production department. 

Kind regards, 

on behalf of

Dr. Johari Yap Abdullah 

Academic Editor

PLOS ONE